# Is the Behavioral Regulation in Exercise Questionnaire a Valid Measure in Older People?

**DOI:** 10.3390/healthcare11202707

**Published:** 2023-10-10

**Authors:** Tommaso Palombi, Fabio Lucidi, Andrea Chirico, Guido Alessandri, Lorenzo Filosa, Simone Tavolucci, Anna M. Borghi, Chiara Fini, Elisa Cavicchiolo, Jessica Pistella, Roberto Baiocco, Fabio Alivernini

**Affiliations:** 1Department of Developmental and Social Psychology, Sapienza University of Rome, 00185 Rome, Italy; fabio.lucidi@uniroma1.it (F.L.); andrea.chirico@uniroma1.it (A.C.); jessica.pistella@uniroma1.it (J.P.); roberto.baiocco@uniroma1.it (R.B.); fabio.alivernini@uniroma1.it (F.A.); 2Department of Psychology, Sapienza University of Rome, 00185 Rome, Italy; guido.alessandri@uniroma1.it (G.A.); lorenzo.filosa@uniroma1.it (L.F.); simone.tavolucci@uniroma1.it (S.T.); 3Department of Dynamic and Clinical Psychology, and Health Studies, Sapienza University of Rome, 00185 Rome, Italy; anna.borghi@uniroma1.it (A.M.B.); chiara.fini@uniroma1.it (C.F.); 4Department of Systems Medicine, Tor Vergata University of Rome, 00133 Rome, Italy; elisa.cavicchiolo@uniroma2.it

**Keywords:** older adults, Self-Determination Theory (SDT), Behavioral Regulation in Exercise Questionnaire (BREQ-3), motivation, physical activity, exercise

## Abstract

Background: Despite the widely recognized benefits of physical activity for preventing physical and cognitive decline during aging, global estimates indicate that most older adults do not achieve the recommended amount of physical activity due to a lack of motivation. The current research examined the validity and psychometric properties of the Behavioral Regulation in Exercise Questionnaire (BREQ-3) among older adults. Based on Self-Determination Theory (SDT), the BREQ-3 stands out as one of the most extensively utilized tools among exercise motivation studies. Methods: A sample of older adults (N = 383; M age = 73.2 years, SD age = 7.2) completed the BREQ-3 and the Godin–Shepard Leisure-Time Physical Activity Questionnaire (GSLTPAQ). Results: Confirmatory factor analyses confirmed the six-factor structure postulated by SDT, showing good fit indices (CFI= 0.95; RMSEA = 0.05; SRMR = 0.04) and supporting the full measurement invariance of the scale across sex and age groups (65 to 74 years; over 75 years). The construct and criterion validity of the BREQ-3 was upheld through the latent correlations between its subscales and their correlations with the GSLTPAQ. Conclusions: We demonstrated for the first time the effectiveness of the BREQ-3 in assessing all forms of behavioral regulation proposed by SDT in older adults, suggesting that older adults similarly interpreted the items across sex and age groups.

## 1. Introduction

Participation in physical activity and exercise among older adults represents a healthy approach to reducing chronic diseases and mortality and improves overall health and quality of life [1]. Nevertheless, the latest global estimates by the World Health Organization (WHO) showed that 1.4 billion adults (27.5% of the world’s adult population) fall short of reaching the prescribed standard of physical activity, which tends to decrease among both women and men as they get older [2].

A systematic review of the literature conducted by Franco and colleagues investigated older people’s perspectives on physical activity to identify and synthesize the barriers and facilitators to physical activity participation [3]. In all, 40% of the studies reviewed reported that low motivation prevented the participation of older people in physical activity and exercise, although they acknowledged the benefits of such activities [3]. Several factors can contribute to a lack of motivation, including low interest or perceived competence in physical activity. These factors can make people unmotivated or insufficiently motivated to be more physically active [4]. Motivation is one of the most crucial variables explaining exercise and physical activity participation among older adults [3].

Nonetheless, there is a substantial lack of evidence in the scientific literature regarding the validity of measures of motivation toward physical activity in older people (i.e., people aged 65 years and above). Older adults tend to spend more time reflecting on their lives and contemplating their values and priorities, and many of them deal with chronic illnesses [5,6], suggesting that they could conceptualize physical activity differently than young adults or adults. Several literature reviews on barriers and motivators toward physical activity among older adults highlighted that factors such as health worries, fear of injury or pain, self-motivation, social support, confidence, and perceived support affected their adherence to and maintenance of physical activity [7,8,9,10].

In addition, there is evidence of sex differences in adopting exercise among older adults. Women perceived their health as being poorer, encountered more barriers to physical activity, and displayed lower self-efficacy for engaging in such activities than men [11]. Conversely, men exhibited lower motivation when it comes to weight loss or enhancing their physical appearance, in contrast to women, who prioritized these motivations [12]. Given these peculiarities, older adults might conceptualize motivation and interpret motivational items in existing measures differently across sex or age groups. Despite several studies assessing motivation in engaging in physical activity, no measures are currently appropriately validated in older adults. Instead, researchers often assume “measurement invariance” [13], a prevalent issue associated with many self-reporting instruments.

In the present paper, we investigated the psychometric proprieties for older people of a widely adopted measure of motivation in engaging in physical activity [14,15,16,17] based on Self-Determination Theory (SDT) [18,19]: the Behavioral Regulation in Exercise Questionnaire (BREQ-3) [4].

### 1.1. Self-Determination Theory (SDT)

Self-Determination Theory (SDT) stands as a frequently theoretical framework used for assessing the influence of motivational factors and of basic psychological needs (i.e., autonomy, competence, and relatedness) on emotional, cognitive, and behavioral outcomes within the context of education [18,19,20,21,22,23,24,25,26], sports [27,28], health [29,30,31], and exercise [32,33,34,35].

The SDT postulates a multidimensional structure of motivation, categorizing diverse regulatory patterns that represent varying degrees of self-determination (the perception of being the source of one’s behavior). Within the SDT framework, a differentiation is made between intrinsic and extrinsic forms of motivation that drive an individual’s behavior. Intrinsic motivation is the most self-determined form of behavior, occurring when individuals participate in activities driven by their inherent interest and enjoyment [36]. Intrinsically motivated individuals experience feelings of enjoyment, the exercise of their abilities, personal accomplishments, and a sense of enthusiasm [4]. Physical activity and exercise can be pursued for the satisfaction of engaging in a challenging activity. Over an individual’s lifespan, intrinsic motivation plays a pivotal role in learning [19]. Several studies have suggested that intrinsic motivation yields beneficial impacts on overall well-being and life satisfaction, persistence, engagement in activities, and performance [4,30,37,38,39,40,41]. On the other hand, extrinsic motivation pertains to activities carried out for instrumental purposes, obtaining a result distinct from the activity itself [19]. The extrinsically motivated behaviors are expressed in four regulations: external regulation, which refers to behaviors carried out to fulfill external requirements or driven by rewards and punishments from an external source [18]; introjected regulation, which refers to behaviors guided by inner rewards such as self-esteem and the avoidance of feelings like anxiety, shame, or guilt [25]; identified regulation, in which personal importance and value are attributed to the behavior, thereby enhancing the willingness to engage in it, even if the activity in itself is perceived as disagreeable; and integrated regulation, when identified regulatory patterns become integrated into an individual’s sense of self [42] and the behavior aligns with the individual’s other interests and principles. These regulatory mechanisms indicate the extent to which behaviors are internalized, demonstrating the transformation of habits into personally endorsed values and self-regulation. This is especially crucial when examining physical activity behavior. During this process, individuals who engage in physical activity or exercise can move from being driven by external and introjected regulations (controlled motivation) to being motivated by identified and integrated regulations (autonomous motivation) [36]. The most autonomous forms of motivation (i.e., identified and integrated motivations) and intrinsic motivation have been recognized as crucial elements in sustaining ongoing commitment to exercise over the long term [43]. The SDT also encompasses the concept of amotivation, which pertains to the absence of intentionality [25,44]. When individuals experience amotivation, regardless of whether it is intrinsic or extrinsic, they perceive a lack of linkage between their behaviors and the ensuing outcomes, leading to difficulties in recognizing any compelling reasons to engage in a particular activity [45]. The SDT continuum is fully explained in Table 1.

### 1.2. The Behavioral Regulation in Exercise Questionnaire (BREQ)

Numerous tools have been created to assess different facets of human motivation in alignment with the SDT framework. The original Behavioral Regulation in Exercise Questionnaire (BREQ), proposed by Mullan et al. [46], was the first attempt to measure the motivation domain, highlighting different forms of regulation according to SDT. Different versions of the BREQ have been developed over the years to measure motivation to exercise according to SDT. The first version of the BREQ [46] measured external, introjected, and identified regulations, along with intrinsic motivation, showing its validity and reliability in several studies [47,48,49,50]. Given the limitation of this version of the BREQ that did not include a measurement of the amotivation factor, Markland and Tobin [51] developed a second version of the questionnaire including four new items. This new measure was called BREQ-2, and it was composed of a 19-item scale in order to measure five factors (amotivation, external, introjected, identified, and intrinsic motivation). The BREQ-2 has been validated in different countries, becoming one of the extensively employed tools in the field of exercise motivation assessment [52,53,54,55,56]. Within the Italian context, the BREQ-2 was translated and validated by Costa et al. [54] in a sample of 576 gym users.

Their study assessed the internal reliability, construct validity, and criterion validity of the BREQ-2 and confirmed the factorial structure of the scale through exploratory factor analysis. Their findings demonstrated the good psychometric properties of the BREQ-2, confirming its utility as a valuable tool for assessing motivation in the exercise domain, also in the Italian context. Although the BREQ-2 presented good psychometric proprieties, it showed an inability to measure integrated regulation, the most autonomous form of extrinsic motivation proposed by the SDT framework. Given this limitation, a third version of the BREQ was developed to include items for measuring integrated regulation, thereby enhancing the comprehension of distinct motivational mechanisms operating within the realm of physical exercise. This new version was called BREQ-3, and it contained 24 items, 4 for each subscale, assessing the whole continuum of motivation according to SDT [4]. The psychometric proprieties of the scale were examined in several studies conducted in different countries [15,17,57], confirming the six-factor structure of the scale and showing acceptable model fit and invariance across sex and age in a sample of adults and young people. Although the 24-item version of the BREQ-3 revealed good psychometric proprieties, a study conducted by Cid et al. [14] reported an unsatisfactory model fit for the 24-item version of the scale. The authors removed 6 items (one for each factor), developing a shorter 18-item version of the BREQ-3 that substantially improved the fit of the model. Within the Italian context, Cavicchiolo et al. [16] analyzed the factorial structure, validity, and reliability of the short version of BREQ-3 in a sample of Italian young people and adults. Their study confirmed the six-factor structure of the scale, showing acceptable model fit and invariance across sex and age.

### 1.3. The Present Study

Although the BREQ-3 has demonstrated its validity as a tool for assessing motivation to engage in physical activity, according to the SDT framework, most studies have validated the instrument in adults and young adults. At present, no research has been conducted to validate any version of the BREQ among the older adult population. The BREQ’s effectiveness in measuring motivation to exercise within older adults remains an unexplored area. Given the increasing life expectancy, it has become essential to gain insights into the life changes associated with aging. The differences between the young-old (aged 60–74) and the old (aged 75 and above) can significantly affect physical, cognitive, and psychosocial aspects. Several studies have highlighted physical and psychological variations between the youngest old and the old [58,59,60] that could lead to disparities in item conceptualization.

In light of this, the main purpose of the current research was to evaluate the psychometric properties and the validity of the 18-item version of the BREQ-3 in a sample of older adults. We hypothesized the six-factor structure of the BREQ-3, as posited by SDT, and full measurement invariance across sex and age groups. Little research has been conducted on measurement invariance among different age categories of the elderly population. In accordance with other studies, we categorized individuals aged 65 to 74 as the youngest-old, while those falling within the 75 to 84 age range were considered to be in the old category [61,62]. In addition, we hypothesized stronger positive correlations between adjacent subscales of the SDT continuum to provide evidence for the construct validity of the BREQ-3. Moreover, we assessed physical activity using the Godin–Shepard Leisure-Time Physical Activity Questionnaire (GSLTPAQ) [63], expecting significant correlations between the six subscales of the SDT continuum. In line with the scientific literature, we hypothesized stronger positive correlations between the most autonomous forms of motivation (i.e., identified, integrated, and intrinsic) and physical activity [64,65].

## 2. Method

### 2.1. Sample and Procedure

The sample in the present study was composed of 383 older adults from different regions of central Italy. The participants’ average age was 73.2 years (SD = 7.2; min = 65, max = 95), with a slightly higher prevalence of females (51.4%). Before data collection, the study protocol was approved by the Ethics Committee of the Sapienza University of Rome. All participants provided informed consent to participate, in which they were informed regarding the overall aim of the study and their rights to anonymity and confidentiality. The online survey was delivered by email and was completed in around 10 min. The dataset used in this study is not accessible to the public, but interested parties can obtain it from the corresponding author upon reasonable request. The exclusion criteria were as follows: non-Italian speaker, current or past neurological disorder or major medical illness (e.g., dementia, traumatic brain injury, schizophrenia, epilepsy, active nausea, vomiting), current psychiatric disorder (e.g., major depression), or a severe sensory or motor deficit that would preclude physical activity or exercise.

### 2.2. Measures

**The Behavioral Regulation Exercise Scale (BREQ-3)** [51,66]. For this study, we used the 18-item Italian version of the BREQ-3, translated and validated by Cavicchiolo et al. (2022) in a sample of Italian young people and adults [16]. The BREQ-3 consists of 18 items rated on a five-point Likert scale, ranging from 0 (“not true for me”) to 4 (“very true for me”). The items were grouped posteriorly into the following six factors (three items per factor): amotivation, external, introjected, identified, integrated regulation, and intrinsic motivation. These factors reflect the motivational continuum of SDT [44]. We used the Italian version of the scale, which in previous validation studies [16] was established as being equivalent to the original by a team of independent judges.

The items of the Italian and English versions of the BREQ-3 are reported in Appendix A and Appendix B, respectively.

**The Godin–Shepard Leisure-Time Physical Activity Questionnaire (GSLTPAQ)** [63] is widely used in measuring Leisure-Time Physical Activity (LTPA). The questionnaire consists of 3 items to investigate the number of times, ranging on a Likert scale from 0 to 15, one engages in mild (minimal effort), moderate (not exhausting), and strenuous (heart beats rapidly) LTPA for at least 20 minutes’ duration in a typical 7-day period. Then, the Metabolic Equivalent of Task (MET) value (i.e., 3, 5, and 9 for mild, moderate, and strenuous intensity, respectively) is multiplied by each frequency score and summed to obtain a leisure score index (LSI) expressed in arbitrary units [63,67].

Sex was coded into two categories, with 0 indicating females and 1 indicating males. Age was grouped into the two categories of young older adults (65 to 74 years) and older adults (over 75 years) in accordance with previous studies [61,62].

### 2.3. Data Analysis

Mplus 8 software, version 1.6 [68], was used to estimate the proposed model consisting of six correlated factors. Confirmatory factor analysis (CFA) was performed using a Maximum Likelihood Robust (MLR) estimator, given the non-normal distribution of some variables. According to the cut-off values for well-fitted models [69], the model’s goodness-of-fit was assessed by employing the following fit indices: Comparative Fit Index (CFI), Root-Mean-Square Error of Approximation (RMSEA), and Standardized Root-Mean-Square Residual (SRMR). To investigate the measurement invariance of the scale across sex and age groups, a hierarchical series of multigroup confirmatory factor analyses (CFAs) was conducted. This involved progressively imposing more stringent equality constraints on the model’s parameters, following guidelines by Van de Schoot, Lugtig, and Hox [70]. In each step of the analysis, the fit of the nested models was compared using the change in CFI values (∆CFI ≤ 0.01) according to Cheung and Rensvold [71]. Finally, Pearson’s correlations were computed between the six subscales of the SDT continuum and the GSLTPAQ to provide evidence for the construct validity of the instrument.

## 3. Results

### 3.1. Descriptive Statistics

The descriptive statistics showed a non-normal univariate distribution for some items related to amotivation and external regulation, which presented a bias to the right; this could be explained by the tendency for the individuals to use the lowest levels of an answer (i.e., zero and one) for the non-autonomous form of motivation. The descriptive statistics for the items and each subscale are reported in Table 2.

### 3.2. Factor Structure of the BREQ-3

The results of the confirmatory factor analysis (CFA) confirmed the six-factor structure for the BREQ-3: χ2 (153) = 2980.527, *p* < 0.001; CFI = 0.95; RMSEA = 0.05; SRMR = 0.04. All of the loadings were statistically significant (*p* < 0.001). The results of the CFA showed that all of the fit indices indicated a good fit of the model with the empirical data [69,72]. Table 3 presents the goodness-of-fit indices for the BREQ-3 model in the current study, along with the findings for other available versions. The standardized factor loadings are presented in Figure 1.

### 3.3. Measurement Invariance across Sex and Age Groups

Table 4 reports the results of the multigroup confirmatory factor analyses (CFAs) conducted separately across sex and age groups. Regarding the multigroup CFAs across sex, when comparing the configural invariance model to the metric invariance model (i.e., the model with equal factor loadings across groups), the difference in CFI between the models was below the cutoff criterion (∆CFI = 0.003), supporting the hypothesis of metric invariance across sex. Furthermore, comparing the metric and scalar invariance models (i.e., the model with equal item intercepts across groups) provided evidence for full scalar invariance of the scales (∆CFI = 0.002). Regarding the multigroup CFAs across age groups, the comparison of the configural invariance model with the metric invariance model and the comparison of the metric invariance model with the scalar invariance model showed no differences in the CFI, demonstrating the full scalar invariance of the scales (∆CFI = 0.000). Overall, the results support the conclusion that the scales exhibit full scalar invariance across sex and age groups.

### 3.4. Correlations between Motivation Subscales and Physical Activity

Table 5 reports McDonald’s omega values for each subscale and a correlations matrix posited by SDT with stronger positive correlations between adjacent subscales than between subscales. In addition, the results showed a statistically significant negative correlation between amotivation and GSLTPAQ and positive correlations between GSLTPAQ and the other subscales, except for external motivation. Overall, these results provide evidence for the construct validity of the BREQ-3.

## 4. Discussion

Extensive research has provided robust evidence that physical inactivity plays a significant role in leading to various chronic diseases and conditions, such as depression, dementia, and the decline of functional abilities associated with aging [1,73,74,75]. Experimental studies have reported that SDT-based interventions can enhance physical activity [76,77], demonstrating that all forms of autonomous regulation predict physical activity participation across different groups and settings [4]. The assessment of motivation seems crucial to understanding the determinants of physical activity, although the scientific literature lacks significant evidence regarding the validity of measures specifically in the older population. Older adults’ needs and motivations may differ significantly from those of adults or young adults, leading to a different conceptualization of the items that could change the motivation structure. In addition, because the life conditions and priorities of the older population differ based on age, it becomes essential to classify older adults into different age categories to properly evaluate the conceptualization of motivation toward physical activity.

In light of this, the purpose of the present study was to investigate the psychometric proprieties of a widely used measure of motivation toward physical activity, the BREQ, providing evidence about the conceptualization and structure of motivation constructs related to older adults and testing its invariance across sex and age groups.

In line with previous validation studies [14,15,16,17], our findings showed that the hypothesized measurement model had a good fit to the data, highlighting the six-factor structure of the scale: amotivation, integrated, identified, introjected, external regulation, and intrinsic motivation.

Measurement invariance was assessed across all subscales of the BREQ-3, indicating that individuals of different sex and age groups similarly interpreted the items. In line with previous research [54], our results demonstrated the full scalar invariance of the scale, suggesting that the BREQ-3 scores can be reliably compared not only across different age groups (i.e., young adults, adults, and older adults) but also in different age subgroups among older people (youngest old and old). Although the scientific literature reported differences in older people’s motivations for and barriers to engaging in physical activity due to their life conditions (e.g., chronic disease) [7,8,9,10], the older adults similarly interpreted the items across sex and age groups.

Moreover, the present study provides evidence for the instrument’s construct validity. The results revealed strong correlations between the different forms of motivation, as predicted by Self-Determination Theory (SDT) and the Godin–Shepard Leisure-Time Physical Activity Questionnaire (GSLTAPQ). Specifically, all forms of autonomous regulation showed a strong correlation with the GSLTAPQ. Although these patterns aligned with the literature, highlighting the relationship between the most autonomous form of regulation and physical activity [4], integrated regulation proved to be the most strongly correlated form of motivation with physical activity. According to SDT, people naturally strive to achieve higher levels of psychological maturity and integration as they age. With time, individuals tend to develop a more authentic and unified sense of self, formed by a combination of interconnected identities based on their own approved and blended preferences, interests, and values [19,78]. As people age, they become more adept at being authentic and regulating their behavior on the basis of self-endorsed motives.

Moreover, the correlation matrix showed stronger positive correlations between adjacent subscales than between further subscales, supporting the quasi-simplex pattern posited by SDT [14,17]. These findings are crucial since construct validity analyses are not often conducted, which can lead to the reporting of biased results [79].

Our study has made a noteworthy contribution to the existing literature, validating the BREQ-3 in older adults for the first time. Measuring the invariance of the BREQ-3 across sex and different age subgroups in older people represents a novel contribution to the scientific literature. The findings support the BREQ-3 as a valid tool for assessing motivational processes based on the SDT framework within the context of exercise among older adults.

Although the results showed the validity of the BREQ-3 in older adults, the present study is not without limitations. Notably, the sample was limited to the Italian population, and to generalize our findings, future studies should be conducted across different countries and cultures.

## 5. Conclusions

The BREQ-3 demonstrated its effectiveness in assessing all forms of behavioral regulation proposed by SDT in older adults. Moreover, our study established the invariance of the BREQ-3 across sex and age subgroups (i.e., youngest old and old), suggesting that older people interpret and conceptualize the different forms of exercise motivation similarly. Gaining insights into the various types of motivation that older adults experience during exercise holds the potential to predict possible outcomes and to help develop interventions to foster motivation to engage in physical activity among the older population.

## Figures and Tables

**Figure 1 healthcare-11-02707-f001:**
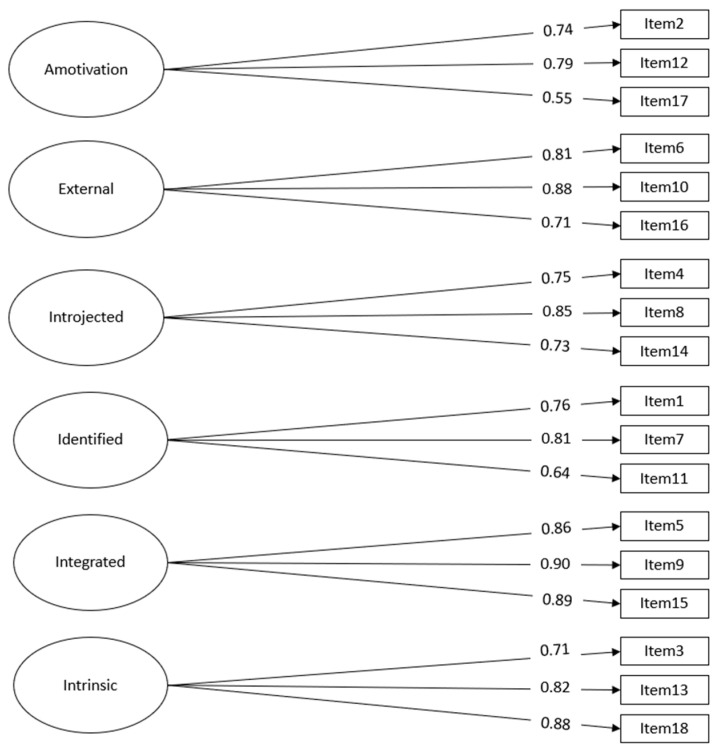
Confirmatory factor analysis. Note: All of the estimates are standardized. All of the estimates are statistically significant at *p* < 0.001. The covariances among latent factors are not reported in this figure.

**Table 1 healthcare-11-02707-t001:** SDT continuum.

	Non-Self-Determined				Self-Determined	
Motivation type	Amotivation	Extrinsic motivation				Intrinsic motivation
Regulatory style	Non-regulation	External	Introjected	Identified	Integrated	Intrinsic
Description	Lack of intentionality in performing exercise	Exercise is performed in order to satisfy an external demand	Exercise is regulated by internal rewards in the form of self-esteem	Personal importance and value is attributed to exercise	Exercise is consistent with one’s life values	People engage in exercise due to their inherent interest and joy

**Table 2 healthcare-11-02707-t002:** Descriptive statistics for the BREQ-3 items and subscales.

						Skewness		Kurtosis	
	N	Mean	SD	Minimum	Maximum	Skewness	SE	Kurtosis	SE
Amotivation	383	0.637	0.813	0	4	1.184	0.125	0.475	0.249
Item2	383	0.809	1.110	0	4	1.316	0.125	0.928	0.249
Item12	383	0.616	0.993	0	4	1.699	0.125	2.247	0.249
Item17	383	0.486	0.909	0	4	2.081	0.125	4.015	0.249
External	383	0.766	0.940	0	4	1.293	0.125	1.161	0.249
Item6	383	0.898	1.161	0	4	1.107	0.125	0.152	0.249
Item10	383	0.872	1.143	0	4	1.099	0.125	0.094	0.249
Item16	383	0.527	0.917	0	4	1.892	0.125	3.189	0.249
Introjected	383	1.627	1.224	0	4	0.191	0.125	−1.131	0.249
Item4	383	1.715	1.447	0	4	0.167	0.125	−1.350	0.249
Item8	383	1.783	1.439	0	4	0.082	0.125	−1.367	0.249
Item14	383	1.381	1.371	0	4	0.501	0.125	−1.099	0.249
Identified	383	2.755	0.974	0	4	−0.670	0.125	−0.249	0.249
Item1	383	2.619	1.163	0	4	−0.586	0.125	−0.421	0.249
Item7	383	2.721	1.327	0	4	−0.765	0.125	−0.641	0.249
Item11	383	2.927	0.989	0	4	−0.814	0.125	0.240	0.249
Integrated	383	1.854	1.315	0	4	0.012	0.125	−1.283	0.249
Item5	383	1.990	1.447	0	4	−0.080	0.125	−1.355	0.249
Item9	383	1.702	1.396	0	4	0.147	0.125	−1.301	0.249
Item15	383	1.869	1.425	0	4	−0.003	0.125	−1.340	0.249
Intrinsic	383	2.179	1.084	0	4	−0.332	0.125	−0.813	0.249
Item3	383	1.687	1.297	0	4	0.076	0.125	−1.179	0.249
Item13	383	2.499	1.155	0	4	−0.606	0.125	−0.405	0.249
Item18	383	2.352	1.263	0	4	−0.452	0.125	−0.828	0.249

Note: SD = Standard Deviation; SE = Standard Error.

**Table 3 healthcare-11-02707-t003:** Goodness-of-fit indices of the BREQ-3 model (including other existing versions).

	N	Mean Age	χ2	df	χ2/df	CFI	RMSEA
English version (BREQ-3) *	207	19.5	357.51	142	2.51	0.92	0.09
English version (BREQ-3) *	132	47.5	253.82	142	1.79	0.93	0.09
Brazilian version (BREQ-3) **	1041	18–60	406.35	215	1.89	0.93	0.07
Spanish version (BREQ-3) ***	524	29.59	689.13	215	3.21	0.91	0.06
Portugese version (BREQ-3) ****	996	23.44	931.69	215	4.33	0.98	0.05
Portugese version (BREQ-3) *****	374	40.51	254.08	120	2.22	0.95	0.06
Italian version (BREQ-3) ******	2222	36.4	833.99	120	6.94	0.96	0.05
Older adults version (present study)	383	73.2	255.81	120	2.13	0.95	0.05

Note: χ2, chi-squared; df = degrees of freedom; χ2/df, normative chi-square; CFI = Comparative Fit Index; RMSEA = Root-Mean-Square Error of Approximation. * [66]; ** [15]; *** [17]; **** [57]; ***** [14]; ****** [16].

**Table 4 healthcare-11-02707-t004:** Goodness-of-fit indices for invariance of the BREQ-3 across sex and age groups.

	χ2	df	χ2/df	CFI	RMSEA	SRMR	Models Compared	∆CFI
Sex (female/male)								
Configural model	477.244	240	1.988	0.941	0.072	0.048		-
Metric model	502.196	252	1.992	0.938	0.072	0.054	metric vs. configural	0.003
Scalar model	521.227	264	1.974	0.936	0.071	0.056	scalar vs. metric	0.002
Age groups (65 to 74; over 75)								
Configural model	492.256	240	2.051	0.937	0.074	0.047		-
Metric model	506.597	252	2.010	0.937	0.073	0.050	metric vs. configural	0.000
Scalar model	516.262	264	1.955	0.937	0.071	0.051	scalar vs. metric	0.000

Note: χ2 = chi-squared; df = degrees of freedom; χ2/df = normative chi-square; CFI = Comparative Fit Index; RMSEA = Root-Mean-Square Error of Approximation; SRMR = Standardized Root-Mean-Square Residual; ∆CFI = difference in the value of the comparative fit index.

**Table 5 healthcare-11-02707-t005:** Correlation matrix between different subscales of the BREQ-3 and GSLTPAQ.

	ω	Amotivation	External	Introjected	Identified	Integrated	Intrinsic	GSLTPAQ
Amotivation	0.74	—						
External	0.84	0.181 ***	—					
Introjected	0.82	−0.252 ***	0.122 *	—				
Identified	0.78	−0.563 ***	0.009	0.512 ***	—			
Integrated	0.91	−0.402 ***	0.060	0.572 ***	0.735 ***	—		
Intrinsic	0.85	−0.441 ***	0.055	0.451 ***	0.730 ***	0.751 ***	—	
GSLTPAQ	0.65	−0.281 ***	0.093	0.234 ***	0.427 ***	0.539 ***	0.428 ***	—

Note: GSLTAPQ = Godin–Shepard Leisure-Time Physical Activity Questionnaire; ω = McDonald’s omega. * *p* < 0.05, *** *p* < 0.001.

## Data Availability

Data are available upon request to the first author.

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
