# Peer review of "Is the Behavioral Regulation in Exercise Questionnaire a Valid Measure in Older People?"

_healthcare, 2023, doi:10.3390/healthcare11202707_

Round 1

Reviewer 1 Report

Line 22 grammatical errors: One of the most extensively utilized tools among exercise....

Line 176 Details about the informed consent and the survey: I don’t believe that the details were given in the manuscript regarding the mode of delivery of the survey. Was this on paper and in-person, over the phone, or online?  There was no mention of a cognitive screen to confirm if the respondents were accurate historians, or were able to self-consent. Given the risk of mild cognitive impairment of this age group, how was cognition screened? If not screened, please acknowledge in the Limitations with details of the importance of cognitive decline, which was mentioned in the abstract and not mentioned again in the paper.

Appendix A: I understand that the survey was provided in Italian but it’s difficult for this reviewer to ascertain the conclusions of the CFA without understanding the content of the Items. Could you please provide an Appendix B with the appropriate translation to English, of the items?

Well done with one required edit in the abstract.

Author Response

R: Line 22 grammatical errors: One of the most extensively utilized tools among exercise....

A: Thank you for your comment. We corrected the grammatical error.

R: Line 176 Details about the informed consent and the survey: I don’t believe that the details were given in the manuscript regarding the mode of delivery of the survey. Was this on paper and in-person, over the phone, or online?  

A: Thank you for your comment. We have added in the manuscript (line 191) the mode of delivery of the questionnaire.

R: There was no mention of a cognitive screen to confirm if the respondents were accurate historians, or were able to self-consent. Given the risk of mild cognitive impairment of this age group, how was cognition screened? If not screened, please acknowledge in the Limitations with details of the importance of cognitive decline, which was mentioned in the abstract and not mentioned again in the paper.

A: Thank you for your comment. We have added sample inclusion and exclusion criteria in the manuscript (lines 193-197).

R: Appendix A: I understand that the survey was provided in Italian but it’s difficult for this reviewer to ascertain the conclusions of the CFA without understanding the content of the Items. Could you please provide an Appendix B with the appropriate translation to English, of the items?

A: Thank you for your comment. We have added the appendix B as you suggested (line 209).

Reviewer 2 Report

I first want to say thank you for allowing me to review this manuscript as it is always a pleasure to review self-determination theory research. The authors did an excellent job on this manuscript and its contents. I have a few suggestions to help improve the manuscript's quality and clarity. 

Section 1.1 SDT - This section is decently robust and gives some background about SDT, however, SDT is more complex than what is presented. Maybe a figure/picture/table would help clarify this for the reader. It gets a little clouded with definitions of terms and such. Additionally, the three basic needs should be mentioned. 

Section 1.2 BREQ - While the history of the BREQ to BREQ-2 to BREQ-3 is informational, I believe this is unnecessary for the scope of the study. Mentioning the development of the BREQ-3 is fine, but no need the depth here (in my opinion). 

Line 173 - 'slightly higher prevalence of females'

Table 2 - I'm not sure where this other data has come from. You have the citations but in my opinion, a table with past study data is more of a review/meta-analysis. I get that this is to compare your study results to others, but this is not the place for this. This information would go into the discussion where comparing and contrasting happens. I just do not like this table with other data outside the study. 

One minor note: change all the 'gender' to 'sex'.

Author Response

R: I first want to say thank you for allowing me to review this manuscript as it is always a pleasure to review self-determination theory research. The authors did an excellent job on this manuscript and its contents. I have a few suggestions to help improve the manuscript's quality and clarity. 

A: Thank you very much for your comment.

R: Section 1.1 SDT - This section is decently robust and gives some background about SDT, however, SDT is more complex than what is presented. Maybe a figure/picture/table would help clarify this for the reader. It gets a little clouded with definitions of terms and such. Additionally, the three basic needs should be mentioned. 

A: Thank you for your comment. We included a new table to clarify the SDT continuum (see table 1). In addition we mentioned the three basic psychological needs posited by SDT (lines 76-77).

R: Section 1.2 BREQ - While the history of the BREQ to BREQ-2 to BREQ-3 is informational, I believe this is unnecessary for the scope of the study. Mentioning the development of the BREQ-3 is fine, but no need the depth here (in my opinion). 

A: Thank you for your comment. We believe that the description of previous versions of the BREQ is important since our study is a validation of this questionnaire among older adults.

R: Line 173 - 'slightly higher prevalence of females'

A: Thank you for your comment. We have edited the sentence in accordance with your comment (line 187).

R: Table 2 - I'm not sure where this other data has come from. You have the citations but in my opinion, a table with past study data is more of a review/meta-analysis. I get that this is to compare your study results to others, but this is not the place for this. This information would go into the discussion where comparing and contrasting happens. I just do not like this table with other data outside the study

A: Thank you for your comment. We believe that your comment is appropriate and agree that the present study is not primarily intended to compare results with other studies. However, we believe it is important to highlight the psychometric properties of the questionnaire both in different cultural contexts and target (e.g. young adults or adults).

R: One minor note: change all the 'gender' to 'sex

A: Thank you for your comment. We changed all the “gender” to “sex”

Reviewer 3 Report

Congratulations to the authors for this interesting paper. 

Some little concerns about it:

-Included Ethical Committee approval number

- I would like to know if before to send the questionnaires it was reviewed by experts (like when you make a new questionnaire)

- Item 20 had > 2.0 and >4.0 in sweetness and kurtosis (amotivation). 

- Why you delete item 8 and 9 and 19? I don't find any explainaition about this items. Please include it and why they were deleted.

- Included the sample calculator (is enough the sample for this study? I consider yes but it could be adequate if it is included)

Thank you

Author Response

R: Congratulations to the authors for this interesting paper. 

A: Thank you very much for your comment.

R: Some little concerns about it:

-Included Ethical Committee approval number

A: Thank you very much for your comment. We included the Ethical Committee approval number in the manuscript (line 373).

R: - I would like to know if before to send the questionnaires it was reviewed by experts (like when you make a new questionnaire)

A: Thank you for your comment. The questionnaire was reviewed by a team of independent judges. We included this information in the manuscript (lines 205-207).

R: - Item 20 had > 2.0 and >4.0 in sweetness and kurtosis (amotivation). 

A: Thank you for your comment. We included an explanation about the non-normality distribution of some items (lines 240-243).

R: - Why you delete item 8 and 9 and 19? I don't find any explainaition about this items. Please include it and why they were deleted.

A: Thank you for your comment. We did not delete any items. We used the reduced version of the questionnaire already validated in Italy by Cavicchiolo et al., as mentioned in the manuscript (lines 199-201). However, we realized that the item numbering was incorrect. We corrected the numbering of the items in Table 2 and figure 1.

R: - Included the sample calculator (is enough the sample for this study? I consider yes but it could be adequate if it is included)

A: Thank you for your comment. The sample size is enough according to various rules-of-thumb that have been advanced in psychometrics including: 1) a minimum sample size of 100 or 200 (Boomsma, 1982, 1985); 2) 5 or 10 observations per estimated parameter (Bentler & Chou, 1987; see also Bollen, 1989); 3) 10 cases per variable (Nunnally, 1967). Furthermore, Table 2 in the manuscript indicates that prior studies conducted the Confirmatory Factor Analysis (CFA) with a smaller sample size.

Reviewer 4 Report

This research investigates the validity and psychometric properties of BREQ-3 among older Italians. This research is an interesting study and can contribute to the literature. But there are a few limitations mostly in the Introduction to be addressed for its publication.

In the introduction, the authors explained the general findings of SDT, BREQ, and BREQ-2, but the authors need to explain more about studies directly investigating the psychometric properties of the BREQ-3 and the limitations of the previous studies related to BREQ-3. The need of the study needs to be proposed based on the limitations of the previous studies closely related to what this study investigates, and in this perspective, the current Introduction is not persuasively described.

The introduction doesn’t sufficiently explain how and why the age groups are proposed. Saying ‘according to other studies’ in line 162 is not enough for authors to use the age groups for this study.

Plus I conceptually easily understand there are stronger positive correlations between adjacent subscales of the SDT continuum to provide evidence for the construct validity of the BREQ-3. But to present it as a hypothesis of this study, the authors need to explain more about why and how this hypothesis needs to be addressed. And there are lack of explanations of the GSLTPAQ in the Introduction. The authors need to present sufficient explanations of the significance and utilization of the GSLTPAQ for this study in the Introduction.

Overall the Method is well written, but the authors need to add explanations of the results of the descriptive statistics, such as normality of the data…. In the table 2, the authors need to disclose the samples of other language versions of BREQ-3, and explain why this study doesn’t report the covariances among latent factors. Plus in the Table 4, GSLTPAQ has higher correlation with Integrated regulation than intrinsic motivation. The authors need to explain that in the Discussion.

Overall The Discussion is well written.

Author Response

R: This research investigates the validity and psychometric properties of BREQ-3 among older Italians. This research is an interesting study and can contribute to the literature. But there are a few limitations mostly in the Introduction to be addressed for its publication. In the introduction, the authors explained the general findings of SDT, BREQ, and BREQ-2, but the authors need to explain more about studies directly investigating the psychometric properties of the BREQ-3 and the limitations of the previous studies related to BREQ-3. The need of the study needs to be proposed based on the limitations of the previous studies closely related to what this study investigates, and in this perspective, the current Introduction is not persuasively described. The introduction doesn’t sufficiently explain how and why the age groups are proposed. Saying ‘according to other studies’ in line 162 is not enough for authors to use the age groups for this study.

A: Thank you for your comment. We added the limitations of the previous studies and explained why we selected these age groups (lines 160-168).

R: Plus I conceptually easily understand there are stronger positive correlations between adjacent subscales of the SDT continuum to provide evidence for the construct validity of the BREQ-3. But to present it as a hypothesis of this study, the authors need to explain more about why and how this hypothesis needs to be addressed. And there are lack of explanations of the GSLTPAQ in the Introduction. The authors need to present sufficient explanations of the significance and utilization of the GSLTPAQ for this study in the Introduction.

A: Thank you for your comment. We reported in the manuscript why the quasi-simplex patterns posited by SDT need to be addressed (see lines 330-331). In addition, we explained our hypothesis about the inclusion of GSLTPAQ in our study (see lines 178-182).

R: Overall the Method is well written, but the authors need to add explanations of the results of the descriptive statistics, such as normality of the data…

A: Thank you for your comment. We included an explanation about the non-normality distribution of some items (lines 240-243).

R: In the table 2, the authors need to disclose the samples of other language versions of BREQ-3, and explain why this study doesn’t report the covariances among latent factors.

A: Thank you for your comment. The covariances among latent factors are not included in the figure but are included in table 5.

R: Plus in the Table 4, GSLTPAQ has higher correlation with Integrated regulation than intrinsic motivation. The authors need to explain that in the Discussion.

A: Thank you for your comment. We included an explanation of this result in the discussion (lines 322-327).